# Sizing SGLT2 Inhibitors Up: From a Molecular to a Morpho-Functional Point of View

**DOI:** 10.3390/ijms241813848

**Published:** 2023-09-08

**Authors:** Silvia Prosperi, Andrea D’Amato, Paolo Severino, Vincenzo Myftari, Sara Monosilio, Ludovica Marchiori, Lucrezia Maria Zagordi, Domenico Filomena, Gianluca Di Pietro, Lucia Ilaria Birtolo, Roberto Badagliacca, Massimo Mancone, Viviana Maestrini, Carmine Dario Vizza

**Affiliations:** Department of Clinical, Internal, Anesthesiology and Cardiovascular Sciences, Sapienza University of Rome, Viale del Policlinico, 155, 00161 Rome, Italy; silviapro@outlook.it (S.P.); paolo.severino@uniroma1.it (P.S.); vincenzo.myftari@gmail.com (V.M.); sara.monosilio@uniroma1.it (S.M.); ludovica.marchiori@gmail.com (L.M.); lucreziam.zagordi@gmail.com (L.M.Z.); domenico.filomena@uniroma1.it (D.F.); gianlucadipietro95@gmail.com (G.D.P.); ilariabirtolo@gmail.com (L.I.B.); roberto.badagliacca@uniroma1.it (R.B.); massimo.mancone@uniroma1.it (M.M.); viviana.maestrini@uniroma1.it (V.M.); dario.vizza@uniroma1.it (C.D.V.)

**Keywords:** heart failure, SGLT2i, cardiac reverse remodeling

## Abstract

Sodium–glucose cotransporter 2 inhibitors (SGLT2i), or gliflozins, have recently been shown to reduce cardiovascular death and hospitalization in patients with heart failure, representing a revolutionary therapeutic tool. The purpose of this review is to explore their multifaceted mechanisms of actions, beyond their known glucose reduction power. The cardioprotective effects of gliflozins seem to be linked to the maintenance of cellular homeostasis and to an action on the main metabolic pathways. They improve the oxygen supply for cardiomyocytes with a considerable impact on both functional and morphological myocardial aspects. Moreover, multiple molecular actions of SGLT2i are being discovered, such as the reduction of both inflammation, oxidative stress and cellular apoptosis, all responsible for myocardial damage. Various studies showed controversial results concerning the role of SGLT2i in reverse cardiac remodeling and the lowering of natriuretic peptides, suggesting that their overall effect has yet to be fully understood. In addition to this, advanced imaging studies evaluating the effect on all four cardiac chambers are lacking. Further studies will be needed to better understand the real impact of their administration, their use in daily practice and how they can contribute to benefits in terms of reverse cardiac remodeling.

## 1. Introduction

Sodium–glucose cotransporter-2 inhibitors (SGLT2i), or gliflozins, are a class of drugs originally developed for patients with type 2 diabetes mellitus (T2DM) and now recommended for the treatment of heart failure (HF) with reduced ejection fraction (HFrEF) and preserved ejection fraction (HFpEF), regardless of blood glucose value, by the latest European and American guidelines on HF [1,2,3,4].

Their mechanism of action is based on the capability to inhibit type 2 sodium-glucose cotransporter, the main isoform expressed in kidneys at the level of the S1 segment epithelial cells of the proximal tubule, which is responsible for the active reabsorption of approximately 90% of the glucose filtered by the renal glomerulus. Indeed, it has been shown that the drug is first filtered by the glomerulus and then inhibits the molecular target directly from the tubular lumen extracellularly [5]. Interestingly, cardiac tissue does not express the SGLT2 receptor; thus, most of the cardiac effects are mainly indirect.

The diuretic effect has been one of the first to be explored, as congestion is a major cause of hospitalization both in HFrEF and HFpEF. Initially, it was thought to be primarily related to the inhibition of glucose reabsorption and, thus, to the osmotic action generated by glycosuria. Lately, several studies demonstrated the efficacy of these drugs regardless of the presence of T2DM in HF patients, and additional action mechanisms began to be brought to light, as SGLT2i drugs possess an important natriuretic effect, which is combined with the ability to increase glucose excretion. It contributes to the reduction of volemia without activating counterregulatory mechanisms, such as the sympathetic nervous system or the renin–angiotensin–aldosterone system (RAAS), which commonly lead to pathological cardiac remodeling [6]. More specifically, gliflozins are also able to reduce the activity of another molecule expressed at the proximal tubule, the type 3 sodium–hydrogen exchanger (NHE3), whose activity is closely related to SGLT2, explaining the natriuretic effect of these drugs [7,8]. SGLT2i also causes interstitial drainage by increasing plasma osmolarity and the gradient between the plasma and the interstitial space, with a lower reduction of the intravascular volume compared to loop diuretics [9].

The decongestive power of SGLT2i has been studied in acute settings [10,11]. SGLT2i, in fact, may indirectly slow edema development with different mechanisms, such as improvement of the stroke volume and reduction of inflammation and endothelial dysfunction, as is further discussed [12,13].

However, all of these biological mechanisms linked to a diuretic effect are not enough to explain the many cardiovascular (CV) benefits of SGLT2i. Therefore, the off-target effects on myocardiocytes must also be considered [14].

The main molecular effects of SGLT2i on cardiovascular circulation will be discussed in depth below and are summarized in Figure 1.

## 2. Molecular Effects of SGLT2i

### 2.1. Electrolytes Homeostasis

Many clinical studies agree on the fact that most of the benefits of SGLT2i on myocardial activity are due to their ability to suppress the cardiac sodium–hydrogen exchanger isoform (NHE1), which has an important modulatory effect on intracellular sodium homeostasis and, subsequently, prevents cellular calcium overload [15]. Indeed, the latter plays a central role in the electromechanical coupling of cardiomyocytes. During systole, the depolarization of the plasmatic membrane activates calcium’s L-channels, resulting in the entry of this ion into the cell. The entry of calcium from outside stimulates the mechanism of calcium-induced calcium release, mediated by the activation of ryanodine receptors. This allows the cell, itself, to release more calcium from the calciosomes, thus reaching a concentration that is high enough to activate muscle contraction. On the other hand, during diastole, repolarization closes the membrane channels on the cell’s surface, and calcium is partly re-uptaken within the sarcoplasmic reticulum by the sarco-endoplasmic reticulum Ca^2+^-ATPase 2a (SERCA2a) pump. The remaining calcium is thrown back out of the cell by the Na^+^/Ca^2+^ exchanger (NCX) and the plasma membrane Ca^2+^-ATPase (PMCA) pump, both expressed on the sarcolemma [16,17]. It has been shown that a large part of the pathological changes in HF is precisely due to a malfunction of these mechanisms; in particular, excessive activity of phospholamban would lead to an increased inhibition of the SERCA2a pump and, thus, an increase in intracellular calcium concentration, resulting in a lengthening of the diastolic phase of the cardiac cycle and a reduction in the availability of calcium in the sarcoplasmic reticulum for the systolic phase [17,18].

The direct effect of the overexpression of SGLT1 and NHE1 on myocardial sarcolemma is excessive sodium accumulation which, in turn, leads to a reduction in the activation of the NCX, with a further increase in intracellular calcium. This causes an increase in pathological mechanisms, mostly oxidative stress, which forces the cell itself to activate the molecular pathways of programmed death [17,19,20]. Thus, the inhibitory influence exerted by SGLT2i on NHE1 can improve the overall function of myocardial tissue by preventing excessive accumulation of sodium and calcium in the cytoplasm and, thus, autophagy as the primary mechanism of cell death [21].

A combination of these mechanisms could improve the clinical status of patients and reduce the risk of death or rehospitalization. Moreover, it can slow down, if not even reverse, the set of processes that lead to pathological cardiac remodeling by improving the supply of oxygen and cellular metabolic system [22].

These drugs act not only on sodium and calcium but also on potassium, as recently suggested. In fact, a further association exists between SGLT2i and a reduced risk of hyperkalemia, a very frequent complication in HF patients that is closely related to comorbidities (i.e., chronic kidney disease and diabetes mellitus), as well as to HF therapy (such as the renin–angiotensin–aldosterone system (RAAS) inhibitors) [23]. In patients with high serum potassium levels, the risk of developing arrhythmias and sudden death is greatly increased. Neuen et al. conducted a meta-analysis incorporating data from six trials, observing a clear reduction in the risk of hyperkalemia in patients taking SGLT2i compared to a placebo (hazard ratio: 0.84 (95% CI 0.76–0.93)) [24]. Moreover, significant results were found in a systematic review, conducted by Yang et al.: drugs such as aldosterone receptor antagonists (MRAs) were associated with an increased risk of hyperkalemia, as well known, while SGLT2i reduced it (RR 2.08, 95% CI 1.86–2.33 vs. RR 0.78, 95% CI 0.65–0.93) [25].

In addition, the intrinsic decongestant mechanism of SGLT2i has allowed for the reduction of the use and dose of loop diuretics in clinical practice, obtaining a great improvement in serum electrolyte homeostasis [26].

### 2.2. Metabolic and Energy Modifications

The heart is clearly one of the most oxygen-consuming organs. Differently from the brain, where neurons are practically forced to use glucose as their only energy substrate to synthesize adenosine triphosphate (ATP) molecules, cardiac cells can adapt their energy source. This adaptation is based on the availability of the substrate, the degree of perfusion and the workload required [17,27]. At rest, the primary source of energy is represented by beta-oxidation of free fatty acids (FFAs). However, it has been shown that glucose uptake by cardiac cells is markedly increased in the postprandial period because of the increased blood glucose and insulin levels [17,28]. During intense physical activity, the heart prefers to conserve its oxygen supply as much as possible, contenting itself with producing ATP through anaerobic glycolysis or by the lactate molecules produced by the skeletal muscle itself. Similarly, when the concentration of ketone bodies in the blood increases, the heart can further change its energy preference, favoring the latter over both FFAs and glucose, as their oxygen demand is much lower, while their energy yield is higher [15,17]. Under pathological conditions, this unique metabolic plasticity of cardiomyocytes is both a source of survival and a source of damage greatly limiting the functionality of the cell. Over the long term, the increased use of glucose in place of FFAs leads to the downregulation of beta-oxidation genes, such as peroxisome proliferator-activated receptor α (PPARα). The consequence is the accumulation of lipids, which will be attacked by free oxygen molecules, as they can no longer be used by the cell as an energy substrate, thus producing reactive oxygen species (ROS) [17,29]. Lipotoxicity, therefore, contributes to an increase in oxidative stress in HF, and, at the same time, the metabolic shift toward glucose oxidation produces a chronic energy deficit that promotes cellular dysfunction [17,29,30]. Many studies have pointed out that SGLT2i can also increase levels of ketone bodies, which are used as the main source of energy by the heart, contributing to an improvement in overall cardiac function. However, the real underlying mechanism is not yet fully clarified. In a study conducted by Wallenius et al. in a population of rats with diabetes, it was shown that the administration of dapaglifozin (1 mg/kg, 4 weeks) increased ketone body levels thanks to the increased mobilization of FFAs from major storage sites, such as adipose tissue, and their beta-oxidation mainly in the liver [31]. A study by Santos-Gallego and colleagues showed that in a population of nondiabetic pigs with ischemic heart disease, artificially induced by balloon-occlusion of the proximal part of the anterior descending artery, empaglifozin was related to a marked improvement in left ventricular (LV) remodeling (lower LV mass, reduced LV dilatation and less LV sphericity) and ejection fraction (EF) after two months of treatment. This study also demonstrated that these results were connected to an increase in consumption of FFAs and ketone bodies compared to the control group in which glucose remained the main energy substrate [32]. Ketosis induced by SGLT2i may be beneficial for the kidneys because of the reduction in oxidative stress and inflammatory status, but the potential risk of inducing diabetic ketoacidosis (DKA) should not be underestimated, mostly in the treatment of patients with type 1 diabetes mellitus (T1DM) [33].

In fact, a meta-analysis of randomized controlled trials showed that SGLT2i increases the risk of DKA significantly in patients with T1DM (RR 4.49, 95% CI 2.88, 6.99) in a dose-dependent way, with a 4.9-fold higher rate at high doses of SGLT2i (34 events per 1000 person-years) than at low doses (7 events per 1000 person-years) [33,34]. The absolute rate of DKA among individuals with T2DM treated with an SGLT2i is much lower (0.6–2.2 events per 1000 person-years), while the absolute rate of DKA among T2DM patients in observational studies fluctuates from 0.6 to 4.9 per 1000 person-years, corresponding to a nonsignificant 1.7-fold higher risk (RR 1.74, 95% CI 1.07, 2.83; *p* = 0.12) [33,35].

This feature makes gliflozins capable of stimulating those pathways that are normally activated by starvation, such as ketogenesis and gluconeogenesis, in addition to the reduction of hyperglycemia [36]. Although the underlying molecular mechanisms have not been fully elucidated yet, a highly supported hypothesis is that the activation of SIRT-1 (nicotinammide adenina dinucleotide-dependent deacetylase sirtuin-1), its effectors PGC-1α (peroxisome proliferator-activated receptor gamma coactivator 1-alpha) and FGF21 (fibroblast growth factor 21) can play a pivotal role. Moreover, the activation of the SIRT1/AMPK (adenosine monophosphate) signaling pathway, which culminates in the inhibition of Akt/mTOR (protein kinase B/mammalian target of rapamycin), fosters a condition of energy deprivation. This shift is accompanied by decreased oxidative stress, restoration of mitochondrial function, lowered inflammation, heightened contractile activity and an upsurge in autophagy [37]. Similarly, promoting through glycosuria a systemic starvation condition, SGLT2i stimulates the activation of SIRT-1 in every cell, reprogramming their energy metabolism, thus promoting the production of ketone bodies [38,39].

However, glycosuria is related to an increased risk of urinary tract infection (UTI), one of the main reasons for the discontinuation of the treatment [40]. The prevalence of UTI among patients with T2DM treated with SGLT2i is approximately 9% [41], and the responsible agents are fungi, bacteria, viruses and parasites. The increased glucose concentration in urine promotes the multiplication of pathogens. Poor genital hygiene and uncontrolled glycemic levels are the main risk factors and must be optimized for an efficient prevention. [42].

### 2.3. Oxidative Damage and Microcirculation Involvement

Cardiac microcirculation also plays a central role in the pathophysiology of HF, especially for HFpEF. Endothelial cells can undergo significant oxidative stress caused by inflammation. In their study, Uthman et al. hypothesized that inflammation activates the NHE1 transporter and, thus, alters the ionic homeostasis of the cell itself by increasing the inner concentration of sodium. This would potentially lead to the increased production of ROS. Starting from this hypothesis, they tested the effectiveness of empaglifozin in blocking this pathological pathway in vitro by inhibiting NHE1 [43]. This exchanger is also activated in diabetic patients with hyperinsulinemia, as well as by the accumulation of epicardial fat, which contributes to increased pro-inflammatory cytokines production on both the myocardium and coronary microcirculation level [44]. Braha et al. tested the relationship between both epicardial fat thickness and epicardial fat total volume with systolic–diastolic dysfunction in a population of 53 patients with T2DM since the commencing of dapaglifozin. In fact, they demonstrated that after 24 weeks, there was a reduction in both epicardial fat total volume (37.8 ± 17.2 vs. 20.7 ± 7 cm^3^; *p* < 0.001) and thickness (5.95 vs. 3.01 mm; *p* < 0.001), with significant improvement in type 1 diastolic dysfunction, while they did not find similarly significant improvements in LVEF [45].

A further protective mechanism against oxidative stress is represented by mitophagy, a selective autophagy that leads to the exclusive degradation of mitochondria damaged by ROS. In fact, both ischemia and reperfusion cause a series of damage to the microcirculation, including increased endothelial permeability and infiltration by neutrophil granulocytes. The combination of these alterations, known as endothelial dysfunction, leads to the increased expression of adhesion molecules for the recall of other inflammatory cells and to the production of molecules with vasoconstrictive action such as endothelin-1. It also causes a reduction in the expression of protective factors, such as cadherins, which are essential for maintaining adhesion between endothelial cells and pericytes, and nitric oxide synthase. In this regard, Cai and colleagues demonstrated, in a population of mice with ischemia/reperfusion injury, through Western blot analysis of mitophagy markers, that empaglifozin can safeguard cardiac microcirculation by minimizing oxidative damage to endothelial cells. Precisely, it is able to stimulate mitophagy by activating the AMPKα1 (AMP-activated protein kinase catalytic subunit alpha 1)/ULK1 (Unc-51-like autophagy-activating kinase)/FUNDC1 (FUN14 domain-containing 1)/mitophagy pathway [46]. Furthermore, a study by Nakao et al. aimed to prove the link between microcirculation and cardiac function and to test how empaglifozin improved capillarization in a population of mice with LV systolic dysfunction induced through transverse aortic constriction. Through a metabolome and transcriptome analysis, they were able to show that empaglifozin administration stimulated the AKT (serine/threonine kinase) pathway resulting in the phosphorylation of endothelial nitric oxide synthase (eNOS) and, thus, nitric oxide (NO) production. The histological results obtained subsequently confirmed the positive effect of the drug on microcirculation. A marked reduction in capillary rarefaction was observed along with the apoptosis of endothelial cells. At the same time, they showed an improvement in systolic dysfunction during LV pressure overload [47].

### 2.4. Serum Biomarkers Effects

Clinically, several studies have confirmed the efficacy of SGLT2i in preventing hospitalization and CV death in both HFrEF and HFpEF patients, regardless of the presence or not of T2DM [48,49,50,51,52].

Thus, an attempt was made to find reliable cardiac biomarkers that could identify which patients would benefit the most from these drugs before treatment began. Patients with the highest amino-terminal pro-b-type natriuretic peptide (NT-proBNP) were supposed to have the highest CV risk reduction [53]. However, it was observed that the reduction in the incidence of major adverse cardiac events (MACEs) was achieved for all patients enrolled in the different studies, regardless of their baseline NT-proBNP value, with no significant differences among the groups [53,54]. Despite this, SGLT2i when chronically used is, indeed, associated with a significant reduction in NT-proBNP. In a study by Januzzi and colleagues, an 11% reduction (*p* < 0.001) in NT-proBNP was already demonstrated at 1 year after canaglifozin administration and an even greater reduction at 6 years (*p* = 0.004), compared to the control group [55]. Januzzi et al. tested empaglifozin on HFrEF patients in who biomarkers were measured at the beginning and then after 4, 12, 52 and 100 weeks. The reduction in NT-proBNP occurred at each timepoint examined, with greater significance at 52 weeks, with a difference from the placebo of 13% (*p* < 0.001) [53]. Finally, dapaglifozin was tested by Butt et al. At 8 months after starting the drug, treatment with dapaglifozin effectively showed at reduction of NT-proBNP levels by 303 pg/mL compared with the group that received placebo (95% CI, −457 to −150 pg/mL) [56]. In the DEFINE-HF trial, the capability of dapaglifozin to improve clinical status was tested, which was evaluated using the Kansas City Cardiomyopathy Questionnaire (KCCQ) and the correlation using the reduction of NT-proBN in patients with HFrEF and NYHA (New York Heart Association) II-III receiving optimal therapy. While there was no significant difference in mean NT-proBNP after 6 and 12 weeks between the dapaglifozin group and the placebo group (1133 pg/dL (95% CI 1036–1238) vs. 1191 pg/dL (95% CI 1089–1304), *p* = 0.43), there were found a mean ≥5-point increase in KCCQ (42.9 vs. 32.5%, adjusted OR 1.73, 95% CI 0.98–3.05) and a ≥20% decrease in NT-proBNP (44.0 vs. 29.4%, adjusted OR 1.9, 95% CI 1.1–3.3) by 12 weeks [57].

Mostly, studies correlating SGLT2i therapy with improvement in bio-humoral assessment are conducted on patients with chronic HF. The EMPA-RESPONSE-AHF study aimed to test the use of empaglifozin against placebo in a population of 80 patients with acute HF. After 30 days of treatment, no significant differences were observed between the two groups in the visual analogue scale, dyspnea score, diuretic response, hospitalization time, or change in NT-proBNP, but they confirmed that SGLT2i are a safe and manageable class of drugs that can increase significantly diuresis compared to the placebo, especially during the first 4 days of hospitalization (difference 3449 (95% CI 578–6321) mL; *p* <  0.01) and the risk of rehospitalization or death at 60 days [10].

## 3. Cardiac Remodeling

Pathological remodeling in HF is a complex process that involves structural and functional changes in the heart in response to various pathophysiological stimuli. It encompasses alterations in cardiac structure, including myocardial hypertrophy, fibrosis, and changes in chamber dimensions [58]. Several factors contribute to the development of pathological remodeling, including neurohormonal activation, inflammation, oxidative stress, and mechanical stress on the myocardium. Indeed, it is a multifactorial process involving intricate molecular, cellular, and physiological interactions. Understanding the underlying mechanisms and identifying key molecular targets involved in cardiac remodeling can pave the way for the development of novel therapeutic interventions in order to attenuate or to reverse the adverse remodeling and to improve outcomes for patients with HF [59].

Considering SGLT2i’s well-known, significant results in reducing CV death and HF hospitalization [1], there is increasing interest regarding the assessment and quantification of cardiac remodeling after SGLT2i administration [60]. In particular, noninvasive morpho-functional study, using mostly echocardiography and cardiac magnetic resonance imaging (CMR), is being applied in numerous trials to assess whether and to what extent SGLT2i can induce reverse cardiac remodeling.

The main effects of SGLT2i on cardiac reverse remodeling will be discussed more in depth below and are summarized in Figure 2.

### 3.1. Left Ventricular Volumes and Left Ventricular Mass

The effects of SGLT2i on LV volumes and mass have been widely explored over the last years through echocardiography and CMR assessment, with discordant results. A recent meta-analysis, including 32 different studies, pointed out a neutral effect of SGLT2i on LV end-diastolic volume (LVEDV) and a significant reduction of the LV end-systolic volume (LVESV) [61].

LV volumes are a strong predictor of adverse CV outcomes in patients with myocardial infarction, and the potential effect of SGLT2i in reducing them is an interesting point to explore [62].

Notably, a significant reduction of LVEDV and LVESV after 12 weeks of treatment emerged from a subanalysis of the EMPIRE HF trial [63]. The same promising results were obtained by the EMPA-TROPISM study, which demonstrated a meaningful improvement of ventricular volumes in patients treated with empagliflozin [64].

However, the DAPA-LVH trial did not show a significant change both in LVEDV and LVESV but rather a reduction of the LV mass (LVM) [65]. Moreover, a meta-analysis by Theophilis et al. showed an overall reduction of LVM in the studies investigated [50]. Chuliàn et al., for instance, demonstrated a greater reduction of LVM if compared to previous studies [66].

LV hypertrophy (LVH) significantly influences CV outcomes and overall mortality rates in the general population [67]. In fact, SGLT2i may be responsible for decreasing the LVM by reducing blood pressure (BP), ventricular preload, and systemic inflammation leading to a reduction of the incidence of all major CV events, including sudden deaths, HF hospitalizations, new onset atrial fibrillation, and strokes. Nevertheless, the effective role of SGLT2i in patients with CV risk factors, other than T2DM, is still to be established, since other studies, such as the EMPA-HEART 2 trial, did not manage to prove a meaningful reduction in LVM after 6 months of treatment with empaglifozin [68]. All in all, the effect of SGLT2i on LVM seems to be greater in patients with greater LVM at baseline [69].

### 3.2. Left Ventricular Systolic Function

LVEF represents the main parameter for the diagnosis of HF patients, tailoring therapeutic management and risk stratification. Its limitations to fully cover the complexity of the HF syndrome are well known, but it still remains the principal parameter in European and American guidelines to classify the different types of HF, even if this can be criticized [70,71,72].

Several trials have explored the role of SGLT2i on left ventricle systolic function, with a particular focus on LVEF and global longitudinal strain (GLS). These studies did not provide definitive results, even if the impact of SGLT2i seems considerable [61,73]. For example, the ADD DAPA trial, revealed a relevant increase in LVEF as the primary endpoint in patients treated with dapaglifozin on top of ARNIs (angiotensin receptor/neprilysin inhibitors). The advantages of adding dapaglifozin to the standard care is enhanced by the hypothesis that the SGLT2i mechanism is independent from the other main HF drugs [74].

GLS is a relatively new parameter for HF patients, since it reflects the function of the heart’s longitudinal fibers. GLS emerged as a valuable parameter over the past few years because of its significant reliability and was included in the guidelines for what concerns the assessment of systolic function [60,75].

As for LVEF, GLS change has been one of the most important endpoints of studies on reverse cardiac remodeling. Ikonomidis et al. found a significant improvement in GLS after 12 months treatment with empaglifozin [76]. Similar findings have been obtained by other groups [66,77]. These effects may be partly explained by the increased myocardial energy supply fostered by SGLT2i administration [32].

However, the promising effects of gliflozins on reverse cardiac remodeling have not always been confirmed, particularly in improving LVEF and GLS. For instance, Rau et al.’s study showed that empaglifozin did not affect LV systolic function, as indicated by unchanged LVEF and GLS values [78]. The same outcomes were shown in other trials [79,80]. Furthermore, no significant changes in GLS were found in the SUGAR-DM-HF [81].

Thus, the role of SGLT2i in increasing systolic function remains unclear, and further studies are needed in order to guarantee the best outcome for patients and a better understanding of the mechanisms behind their demonstrated benefits in HF.

### 3.3. Left Atrial Volume

Left atrial volume (LAV) plays an essential role in HF, as an indicator of the prognosis and the progression of the disease. An enlarged left atrium is a sign of atrial remodeling, which can lead to atrial fibrillation (AF), further exacerbating HF. Monitoring and managing LAV is essential in the comprehensive care of HF patients, as it provides valuable information for risk stratification, treatment optimization, and prognosis. It is reasonable to explore, therefore, the impact of SGLT2i treatment on LAV reduction.

The DAPACARD trial evaluated LAV after 6 weeks of treatment with dapaglifozin resulting in a significant reduction of LAV [82]. The DAPA-MODA study, in addition, provides evidence on the association between dapagliflozin and the reduction in LAV, which is the main marker of LA remodeling and dysfunction. SGLT2i may be responsible for LA reverse remodeling, namely the process that leads to a reduction in both LAV and the restoration of specific functional parameters [83]. On the other hand, there was no significant difference in atrial volume measures in the CardioLink 6 Trial, which explored using CMR the effects of 6 months of treatment with empaglifozin [84]. The IDDIA trial, moreover, did not provide significant results on LAV decrease. However, it is important to underline that this trial included patients with DM; regardless, they had a history of HF, suggesting that in these patients the benefit on cardiac remodeling may be lower due to their lower cardiac impairment [85].

Beyond the impact on LAV, other studies were meant to evaluate the effect of SGLT2i on AF onset. For instance, from Wang et al.’s meta-analysis, it emerged that the incidence of AF in patients treated with SGLT2i was significantly lower, deepening the current knowledge on the indirect antiarrhythmic effect of glyphozines [86].

### 3.4. Diastolic Function

SGLT-2 has been shown to impact the patient’s prognosis regardless of LVEF [50,87]. In HFpEF patients, diastolic function plays a fundamental role [88,89], even if it is a very complex concept that has been revised over the recent years. Currently, the presence of diastolic dysfunction alone does not imply the presence of clinical HFpEF [88,90,91], since other parameters must be considered, such as the presence of mild systolic LV dysfunction, the size and function of the LA, the increase in vascular rigidity, the coronary and microcirculation dysfunction and abnormalities of right ventricular–pulmonary coupling [89]. Although the ESC Guidelines recommend using different diastolic dysfunction indices in conjunction with BNP dosage [1], this approach is deficient in sensitivity [92]. In multivariable regression studies, only two echocardiographic variables were found to significantly predict HFpEF: septal E/e’ > 9 and RVSP > 35 mmHg [92].

However, considering all of the multiple characteristics of these drugs, SGLT2i appear to be molecules capable of acting on diastolic dysfunction from different points of view.

The effects of SGLT2i on diastole have been studied for years. A study by Pabel et al. analyzed myocardial trabeculae isolated from advanced HF patients [93]. Empagliflozin administration significantly reduced abnormally increased diastolic tension in HF hearts, without reducing systolic tension [93]. In addition, the same study showed that empaglifozin improved the wall tension without affecting the calcium cycle [93,94] both in human and animal hearts but acting on the phosphorylation of myfilamentary proteins, which is generally reduced in the case of diastolic dysfunction [93,95].

The E/e ratio is significantly reduced after SGLT2i therapy [61,96] from the latest meta-analysis.

In a study by Marketou et al., dapagliflozin showed a significant improvement in LV diastolic dysfunction, evaluated using diastolic stress echocardiography, compared to the placebo with a significant reduction in E/e during exercise [61,97].

The same result was shown in the EMPA-REG OUTCOME study. It showed a rapid and progressive reduction in the E/e ratio without affecting other hemodynamic parameters, such as the cardiac index [61,78,97].

The STADIA-HFpEF and IDDIA trial also confirmed the beneficial effect of SGLT2i, both empaglifozin and dapaglifozin, on diastolic function [61,85,97,98].

Also canaglifozin showed a significantly decrease in the E/e ratio after treatment, especially when the patients also experienced a gain in hemoglobin levels [99].

On the contrary, in the Soga at el. study, even though there were no significative changes in the E/e ratio after 6 months of dapagliflozin therapy, the difference was bigger in patients with dyslipidemia [100].

Given the complexity of diastolic function and also its interconnections with systolic function, Tanaka et al. demonstrated that the improvements in GLS and diastolic function after dapaglifozin administration were strictly connected [101].

The encouraging results of the effects of dapaglifozin on hemodynamic changes during exercise are derived from the randomized trial of Kayano et. al, which showed an early and significant decrease in RVSP during exercise and E/e ratio after therapy with dapaglifozin, while not afflicting the cardiac index [102].

However, the real effect of SGLT2i on diastolic dysfunction is of debate. For example, Prasad et al.’s meta-analysis concluded that there were no significant changes in diastolic or systolic function before and after dapaglifozin therapy against a substantial improvement in LV stroke volume [103]. In a retrospective study by Swathy et al., after one year of treatment with gliflozin, no effect of improvement of diastolic dysfunction was shown, but, on the contrary, there was a significant increase in LV mass [104].

### 3.5. Right Heart Remodeling

The efficacy of SGLT2i on improving the function and morphological remodeling of the left heart has been widely addressed in the literature. However, the same cannot be said of the right heart given the lack of studies in this regard.

Several studies have shown the presence of RV systolic and diastolic dysfunction in T2DM patients with HFpEF. Hence, there is a growing interest in evaluating the possible impact of SGLT2i added to optimized therapy on the right heart, both on morphology and function.

For example, Tadic et al. demonstrated the presence of a higher deformation of the RV through a series of two- and three-dimensional echocardiographic screening in patients with T2DM and prediabetes compared to controls (3DE RV end-diastolic volume index: 69 ± 10 vs. 63 ± 8 vs. 58 ± 8 mL/m^2^, *p* < 0.001; 3DE RV end-systolic volume index: 25 ± 4 vs. 23 ± 4 vs. 22 ± 4 mL/m^2^, *p* < 0.001), with no significant difference in the three groups regarding the 3DE RV ejection fraction (63 ± 4% vs. 62 ± 4% vs. 61 ± 5%, *p* = 0.063) [105]. A further step forward was taken by Gorter’s group, who studied the relationship between RV dysfunction and HFpEF in diabetic patients. A total of 91 patients underwent cardiac catheterization and echocardiographic examination, with a higher rate in the T2DM group (37%) of pulmonary artery hypertension and right systolic and diastolic dysfunction [106]. Recently, a study demonstrated how the addition of SGLT2i (empaglifozin or dapaglifozin) to HFrEF therapy improved RV systolic dysfunction compared to the group without SGLT2i therapy. The first positive changes compared with the baseline values were visible after 3 months of follow-up: RV longitudinal strain of the RV free wall (+7.2%, *p* < 0.001), RV ejection fraction (+10.1%, *p* = 0.003), tricuspid annular plane systolic excursion (TAPSE) (+4.5 mm, *p* = 0.002), s’ wave (+3.5 cm/s, *p* = 0.032), and FAC (+9.0%, *p* = 0.029) [107].

### 3.6. Hemodynamic Forces

During the phases of the cardiac cycle, the ventricles apply forces on the blood generating directional changes, because of the cyclic alternation of contraction and relaxation, known as hemodynamic forces [108]. These forces alter blood flow by creating pressure gradients and accelerations. Subjected to hemodynamic forces, the blood exerts a “hemodynamic response” on the walls of the myocardium that can affect the function and morphology of the myocardium itself [109]. The intensity of the expressed force and its duration reflect the characteristics of the cardiac chamber; in fact, LV does not express the same hemodynamic forces as RV [110] and, at the same time, a performing heart does not express the same forces of a dysfunctional one [111].

Given the counter-effect of forces themselves on the myocardium, many authors have, therefore, speculated that these hemodynamic forces are a result of increased wall tension and pathological remodeling [108].

Hemodynamic forces can be analyzed by echocardiographic examination, both 2D and 3D, and this evaluation allows for highlighting very early abnormalities involving the deterioration of heart function [105].

The quantification of the hemodynamic force of LV and RV has been verified in terms of reproducibility and accuracy, also with 4D flow CMR, allowing for the development of accurate software for their quantification [112].

Although hemodynamic forces are beginning to find application in HF diagnosis and therapeutic success evaluation [113,114], studies showing changes in these strains after therapy with SGLT2i are currently missing.

### 3.7. Myocardial Work

Advanced echocardiography also allows for the reconstruction of the volume–pressure curves of the cardiac cycle. From this reconstruction, a series of “advanced parameters” can be calculated, including the myocardial work index (MWI), myocardial work efficiency (MWE) and myocardial wasted work (MWW) [115,116]. The Ikonomidis et al. study found that all patients after 4 months of therapy with SGLT2i and/or glucagon peptide like-1 showed a higher value of MWI and lower MWW [76]. Palmiero et al. demonstrated a significant increase in MWI beyond LVEF, GLS and TAPSE in patients with HFrEF and T2DM after 6 months of treatment with SGLT2i [117]. In addition, Russo et al.’s study explored the HFpEF category, showing a significant augmentation of LVEF, GLS and MWE and a decrease in MWW, also in this phenotype, after SGLT2i treatment [118]. Further advanced echocardiography studies for the evaluation of antiremodeling effects after therapy with SGLT2i are necessary.

## 4. Conclusions

SGLT2i have recently become a key element in the therapeutic management of HFrEF patients and have shown equally promising results in HFpEF. In this review, in addition to the well-known glycosuric effect, we analyzed the role of gliflozins starting from microcellular effects, such as the ability to control intra and extracellular homeostasis and to optimize the main energy pathways of the myocardium, but also in reducing inflammation, oxidative stress and endothelial dysfunction.

Given all these beneficial effects and others, it is not surprising that SGLT2i has shown a positive impact on reverse heart remodeling. In fact, they are associated with a significant reduction in the volumes of LV and LAV and a significant improvement in LVEF and diastolic dysfunction, posing as being responsible for a comprehensive cardiovascular remodeling, as reported in Table 1. However, not all studies showed unequivocal results. Therefore, it seems reasonable to implement a number of large-scale studies to evaluate the effects on the main parameters of cardiac remodeling, increasing data on advanced imaging and right ventricle analysis and deepening the way they act at the microcellular level.

## Figures and Tables

**Figure 1 ijms-24-13848-f001:**
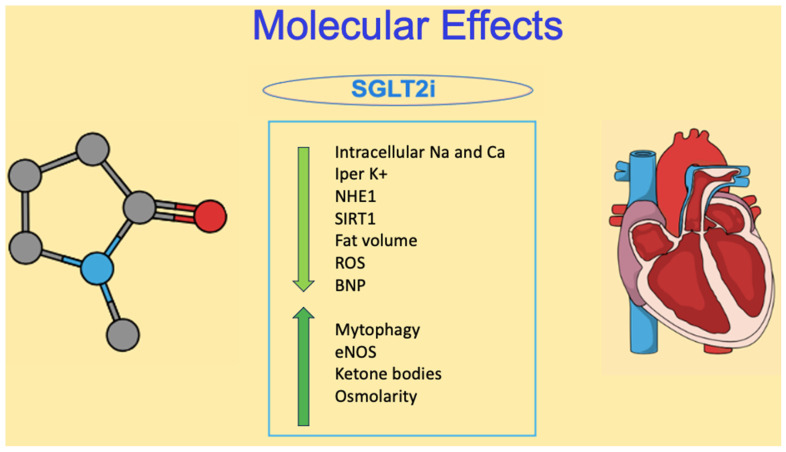
Summary of the main positive effects of SGLT2i drugs on the cardiovascular system. BNP: Pro-b-type natriuretic peptide; Ca: calcium; eNOS: endothelial nitric oxide synthase; K: potassium; Na: sodium; NHE1: sodium–hydrogen exchanger isoform 1; ROS: reactive oxygen species; SIRT1: nicotinammide adenina dinucleotide-dependent deacetylase sirtuin-1.

**Figure 2 ijms-24-13848-f002:**
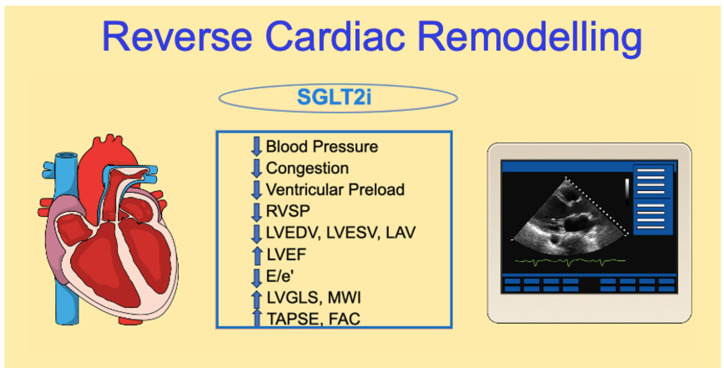
Summary of the main positive effects of SGLT2i drugs on heart reverse remodeling: from hemodynamics to imaging. E/e’: Ratio of early transmitral velocity to tissue Doppler mitral annular early diastolic velocity; FAC: fractional area change; LAV: left atrium volume; LVEDV: left ventricular end-diastolic volume; LVEF: left ventricular ejection fraction; LVESV: left ventricular end-systolic volume; LVGLS: left ventricular global longitudinal strain; MWI: myocardial work index; RVSP: right ventricular systolic pressure; TAPSE: tricuspid annular plane systolic excursion.

**Table 1 ijms-24-13848-t001:** A synthesis of the main SGLT2i CV effects, from bench to bedside.

SGLT2i Effects	Key Points	Reference
SGLT2i and electrolytes	Inhibition of NHE1 by SGLT2i reduced the risk of cellular sodium and calcium overload and improves cardiac function (+)	[21]
	A meta-analysis showed a reduced risk of hyperkalemia in patients on SGLT2i therapy compared to the placebo group (+)	[24]
SGLT2i and metabolism	Empaglifozin switched myocardial fuel metabolism away from glucose toward ketone bodies (+)	[32]
	Dapaglifozin induced hepatic gluconeogenic enzymes expression in obese rats (+)	[36]
	Canaglifozin triggered a fasting-like transcriptional and metabolic program through FGF21 pathway (+)	[38]
SGLT2i and microcirculation	Empagliflozin lowered ROS through NHE-1 inhibition and cellular sodium lowering (+)	[43]
	Empaglifozin activated the mitophagy through the AMPKα1/ULK1/FUNDC1 reducing microvascular damages (+)	[46]
	Empaglifozin reduced capillary rarefaction through the AKT/eNOS/NO pathway (+)	[47]
SGLT2i and NT-proBNP/KCCQ	Empaglifozin reduced NT-proBNP after 52 weeks by 13% compared to placebo (+)	[53]
	Dapaglifozin showed clinical benefits irrespective of baseline NT-proBNP concentration (+)	[54]
	Canaglifozin significantly reduced NT-proBNP already after one year of treatment (+)	[55]
	Dapaglifozin reduced NT-proBNP after 8 months of treatment (+)	[56]
	Dapaglifozin was associated with an improvement in clinical status (≥5-point increase in KCCQ score) (+)	[57]
	Empaglifozin reduced the risk of rehospitalization or death at 60 days in patients with acute decompensated HF (+)	[10]
SGLT2i and LV volumes	A substudy of the EMPIRE HF trial showed a significant reduction of LVEDV and LVESV after 12 weeks with empaglifozin (+)	[63]
	The DAPA-LVH trial did not show a significant change in LVEDV and LVESV (−)	[65]
SGLT2i and LV mass	Dapaglifozin significantly reduced LVM in patients with T2DM and LVH compared to placebo (+)	[65]
	Empaglifozin did not prove a meaningful reduction in LVM in patients without T2DM after 6 months of treatment (−)	[68]
SGLT2i and LVSF	A meta-analysis of RCTs showed that empaglifozin improves LVEF in HF patients compared to the control group (+)	[73]
	Empaglifozin did not change LVSF in terms of LVEF and GLS values (−)	[78]
SGLT2i and LV diastolic function	Dapaglifozin improved left ventricular diastolic function with a significant reduction of E/e’ during exercise (+)	[96]
	A meta-analysis did not show significant changes in diastolic and systolic function after dapaglifozin therapy (−)	[100]
SGLT2i and right ventricle	Therapy with SGT2i in patients with HFrEF improved right ventricular systolic function (+)	[107]
SGLT2i and advanced imaging	Therapy with SGLT2i was associated with an increase in MWI and a lower value of MWW after 4 months (+)	[73]

AKT: Serine/threonine kinase; AMPKα1: adenosine monophosphate-activated protein kinase catalytic subunit alpha 1; DAPA-LVH: dapaglifozin on left ventricular hypertrophy; E/e’: ratio of early transmitral velocity to tissue Doppler mitral annular early diastolic velocity; EMPIRE HF: empaglifozin in heart failure patients with reduced ejection function; eNOS: endothelial nitric oxide synthase; FGF21: fibroblast growth factor 21; FUNDC1: FUN14 domain-containing 1; GLP-1: glucagon-like peptide 1; GLS: global longitudinal strain; HF: heart failure; HFrEF: heart failure with reduced ejection fraction; KCCQ: Kansas City Cardiomyopathy Questionnaire; LV: left ventricular; LVEDV: left ventricular end-diastolic volume; LVEF: left ventricular ejection fraction; LVESV: left ventricular end-systolic volume; LVH: left ventricular hypertrophy; LVM: left ventricular mass; LVSF: left ventricular systolic function; MWI: Myocardial Work Index; MWW: myocardial wasted work; NHE-1: sodium/hydrogen exchanger-1; NO: nitric oxide; NT-proBNP: amino-terminal pro-B-type natriuretic peptide; RCTs: randomized clinical trials; ROS: reactive oxygen species; SGLT2i: sodium–glucose cotransporter-2 inhibitors; T2DM: type 2 diabetes mellitus; ULK1: Unc-51-like autophagy-activating kinase; (+): positive effect; (*−*): nonsignificant positive effects.

## Data Availability

No new data were created.

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
