# Peer review of "Sizing SGLT2 Inhibitors Up: From a Molecular to a Morpho-Functional Point of View"

_ijms, 2023, doi:10.3390/ijms241813848_

Round 1

Reviewer 1 Report

The review by Prosperi et al. is a well organised state of the art review on gliflozins mechanisms of action and clinical effects. The paper is overall well written and my comments are mostly minor:

- in the abstract section, the following sentence should be rephrased "Although SGLT2i appeared to play a positive role in reverse cardiac remodeling and serum cardiac biomarkers". 

- I would mention the Akt/mTOR signal transduction pathway (see doi: 10.1093/eurheartj/suab094) as another possible way gliflozins, by activation of sirtuin 1, exerts cardioprotective properties.  

- Authors mentioned the DKA adverse event in SGLT2i users. However, it should also be mentioned the risk of urogenital infections due to glycosuria.  

- In Table 1 I would divide positive and negative results in two columns or by "+" and "-" symbols instead of bullet points, for the sake of readability and clearness.

- I would add a Figure summarising SGLT2i molecular mechanisms of action.

Reviewer 2 Report

I read with great interest the article titled "Sizing SGLT2 inhibitors up: from a molecular to a morphofunctional point of view" by Prosperi et al.

The paper's design is sound, and the article is logically organized into appropriate sections and subsections. English is generally fine, only minor spell check needed.

Here are the comments and suggested revisions:

1.      About global longitudinal strain there are also recent evidence in well-controlled TDM2 patients with preserved left ventricular ejection fraction who are treated with a SGLT2-i. In these patients a favourable cardiac remodelling, characterized by the improvement of LV-GLS and MWE was observed (doi: 10.1097/FJC.0000000000001450). Please report it.

2.      In addition, in another study on type 2 diabetes and heart failure, despite the lack of a favorable effect on cardiac remodeling, SGLT-2i treatment significantly improved LV systolic and diastolic function, left atrial reservoir and total emptying function, RV systolic function and pulmonary artery pressure (doi: 10.1016/j.diabres.2023.110686). Please report it.

3.       I suggest the authors should produce two figure to make it easy to understand for the readers the role of SGLT2is in cardiac remodeling and molecular effects of SGLT2i.

English is fine.

Reviewer 3 Report

The current review attempted to explore the mechanisms of action of sodium-glucose cotransporter 2 inhibitors (SGLT2i) that have been shown to reduce cardiovascular death and hospitalization across a broad spectrum of patients with HF (HF) and recommended in Class I for the treatment of HF regardless of ejection fraction.

By that date, many comprehensive reviews had been published on the targeted subject – the potential mechanism of action of SGLT2i in HF.

Although the current review is somehow adding to the subject and potentially could benefit readers, it failed to recognize or even mention the effects of SGLT2i on edema or congestion that are the hallmark of HF and major causes of hospitalization/re-hospitalization – both in HFrEF and HFpEF patients.

  1. Hernandez M, Sullivan RD, McCune ME, et al. Sodium-Glucose Cotransporter-2 Inhibitors Improve Heart Failure with Reduced Ejection Fraction Outcomes by Reducing Edema and Congestion. Diagnostics. 2022; 12(4):989. https://doi.org/10.3390/diagnostics12040989

2.       Sullivan RD, McCune ME, Hernandez M, et al. Suppression of Cardiogenic Edema with Sodium–Glucose Cotransporter-2 Inhibitors in Heart Failure with Reduced Ejection Fraction: Mechanisms and Insights from Pre-Clinical Studies. Biomedicines2022; 10(8):2016. https://doi.org/10.3390/biomedicines10082016

3.       Biegus J, Fudim M, Salah HM, Heerspink HJL, Voors AA, Ponikowski P. Sodium-glucose cotransporter-2 inhibitors in heart failure: Potential decongestive mechanisms and current clinical studies. Eur J Heart Fail. 2023 Jul 21. doi: 10.1002/ejhf.2967. Epub ahead of print. PMID: 37477086. 

The authors should have noted in the Introduction that SGLT2 inhibitors were recently added to the American and European guidelines for treating HFpEF.

Authors need to address that the effects of SGLT2i on cardiac tissue are primarily indirect as cardiac tissue does not express SGLT2i receptor SGLT2.

Round 2

Reviewer 2 Report

I have no more comments. The authors appropriately answered to all the issues I raised.

Reviewer 3 Report

The authors have responded well to the previous critiques.